# Expanding the Therapeutic Window of EGFR-Targeted PE24 Immunotoxin for EGFR-Overexpressing Cancers by Tailoring the EGFR Binding Affinity

**DOI:** 10.3390/ijms232415820

**Published:** 2022-12-13

**Authors:** Sei-Yong Jun, Dae-Seong Kim, Yong-Sung Kim

**Affiliations:** 1Department of Molecular Science and Technology, Ajou University, Suwon 16499, Republic of Korea; 2Department of Allergy and Clinical Immunology, Ajou University School of Medicine, Suwon 16499, Republic of Korea

**Keywords:** immunotoxin, anti-EGFR monobody, PE24 toxin, affinity variants, on-target/off-tumor toxicity, therapeutic index, therapeutic window

## Abstract

Immunotoxins (ITs), which are toxin-fused tumor antigen-specific antibody chimeric proteins, have been developed to selectively kill targeted cancer cells. The epidermal growth factor receptor (EGFR) is an attractive target for the development of anti-EGFR ITs against solid tumors due to its overexpression on the cell surface of various solid tumors. However, the low basal level expression of EGFR in normal tissue cells can cause undesirable on-target/off-tumor toxicity and reduce the therapeutic window of anti-EGFR ITs. Here, based on an anti-EGFR monobody with cross-reactivity to both human and murine EGFR, we developed a strategy to tailor the anti-EGFR affinity of the monobody-based ITs carrying a 24-kDa fragment of *Pseudomonas* exotoxin A (PE24), termed ER-PE24, to distinguish tumors that overexpress EGFR from normal tissues. Five variants of ER-PE24 were generated with different EGFR affinities (*K*_D_ ≈ 0.24 nM to 104 nM), showing comparable binding activity for both human and murine EGFR. ER/0.2-PE24 with the highest affinity (*K*_D_ ≈ 0.24 nM) exhibited a narrow therapeutic window of 19 pM to 93 pM, whereas ER/21-PE24 with an intermediate affinity (*K*_D_ ≈ 21 nM) showed a much broader therapeutic window of 73 pM to 1.5 nM in in vitro cytotoxic assays using tumor model cell lines. In EGFR-overexpressing tumor xenograft mouse models, the maximum tolerated dose (MTD) of intravenous injection of ER/21-PE24 was found to be 0.4 mg/kg, which was fourfold higher than the MTD (0.1 mg/kg) of ER/0.2-PE24. Our study provides a strategy for the development of IT targeting tumor overexpressed antigens with basal expression in broad normal tissues by tailoring tumor antigen affinities.

## 1. Introduction

Immunotoxin (IT) is a synthetic chimeric protein consisting of a potent cytotoxic protein toxin linked to a targeting moiety, such as an antibody fragment or ligand, which is specific to an internalizing receptor on cancer cells [1]. Target receptor-mediated endocytosis causes the internalization of IT, which facilitates the cytosolic delivery of toxins to kill cancer cells [2]. Immunogenicity due to the non-human origin of the toxin and dose-limiting toxicity due to the expression of cancer targets in normal cells and tissues pose as primary challenges in the clinical development of ITs [3]. Three immunotoxins have been clinically approved for the treatment of hematologic malignancies, including Denileukin diftitox (Ontak^®^, interleukin-2 (IL-2)- diphtheria toxin) for cutaneous T-cell lymphoma, Tagraxofusp (Elzonris^®^, IL3-DT) for blastic plasmacytoid-cell neoplasms, and Moxetumomab pasudotox (Lumoxiti^®^, anti-CD22 disulfide-stabilized variable fragment (dsFv)-38-kDa fragment of *Pseudomonas aeruginosa* exotoxin A (PE38) toxin) for hairy cell leukemia [1,3]. However, to date, no IT has been approved for clinical use in solid tumor therapy.

Epidermal growth factor receptor (EGFR), a cell surface receptor tyrosine kinase, is commonly overexpressed in a variety of solid tumors, including colorectal cancers and non-small cell lung cancers, and is an attractive anticancer therapeutic target against solid tumors [4]. In this regard, several anti-EGFR antibodies have been clinically approved for the treatment of solid cancers overexpressing EGFR [4]. However, the EGFR-targeting antibodies are not effective in many cancers due to mutations in the *EGFR* and *KRAS* genes, and/or other intracellular signaling pathways [4,5]. The ITs which target EGFR may be alternatives to anti-EGFR antibodies and can deliver the toxin moiety with diverse mechanisms of action to kill cancer cells and overcome cancer resistance [2,6]. In this context, several anti-EGFR ITs, including transforming growth factor alpha-fused PE40 toxin (TGFα-PE40) [7], EGF ligand-fused PE40 toxin (EGF-PE40) [6], anti-EGFR cetuximab-derived scFv-fused PE toxin [8], cetuximab IgG antibody-fused Cucurmosin toxin (C-CUS245C) [9], and anti-EGFR scFv antibody-fused diphtheria toxin (hDT806) [10], have been generated, but none of them have been approved for clinical use so far [2]. The decreased expression of EGFR on the normal epithelium such as the skin and gastrointestinal tract results in on-target/off-tumor toxicity and lowers the dose of ITs available to target tumors, contributing to challenges in its applicability [2,3]. As a result, the therapeutic index, which is the ratio of IT doses that cause toxic versus therapeutic effects, or the therapeutic window, which is the range of IT dosages that can effectively treat cancer without exerting toxic effects, of anti-EGFR ITs should be improved further [11]. The therapeutic window can be expanded by increasing the selectivity of IT for tumors over normal tissues that may also express the antigen [12].

This study aimed to increase the selectivity of anti-EGFR ITs for cells with a higher expression of EGFR compared to cells with low EGFR expression to minimize the risk of on-target/off-tumor toxicity and to expand the therapeutic window of EGFR targeting ITs. As a model toxin, we used a de-immunized 24-kDa PE fragment variant, LO10 PE24 [13,14], which contains the furin cleavage site in domain II (residues 274–284) and the domain III (395–613) of PE with a replacement of the C-terminal endoplasmic reticulum (ER) retention sequence ^609^REDLK^613^ with ^609^KDEL^612^. When PE24 is internalized into endosomes by its targeting moiety, it escapes into the cytosol through the retrograde transport system and inactivates eukaryotic elongation factor 2 by catalyzing adenosine diphosphate (ADP) ribosylation, causing the inhibition of protein synthesis and resulting in cell death [13,14]. We generated a series of the monobody-fused PE24, designated ER-PE24 ITs, with anti-EGFR affinity varying over three orders of magnitude (from *K*_D_ ≈ 0.24 nM to 104 nM), based on a fibronectin-based anti-EGFR monobody with cross-reactivity with both human and murine EGFR [15]. Furthermore, using an EGFR high-expressing A431 cell line and EGFR low-expressing HT29 cell line, we determined the therapeutic window of ER-PE24 ITs in an in vitro cytotoxicity assay. The cross-reactivity of the ER-PE24 variants with both human and murine EGFR facilitated the comparison of the antitumor efficacy and systemic toxicity of ER-PE24 ITs with the highest and intermediate affinity for EGFR in the EGFR-overexpressing tumor xenograft mouse model. This study offers a strategy to expand the therapeutic window of ER-PE24 IT for EGFR-overexpressing cancers by tailoring the EGFR binding affinity.

## 2. Results

### 2.1. Construction of ER-PE24 ITs with Different Affinities for EGFR

We used an EGFR-specific monobody, derived from the tenth type III domain of human fibronectin (Fn3) [15], as an EGFR-targeting moiety to trigger cellular internalization upon binding to EGFR expressed on the cell surface and to facilitate the intracellular delivery of its payload [16]. The anti-EGFR monobody cross-reacts with both human and murine EGFR [16], enabling us to evaluate antitumor activity and systemic toxicity in mouse models. To construct a series of monobody variants with different affinities for EGFR, we selected six residues (Tyr29, Gln30, Thr49, Val54, Lys79, Tyr83), which are known as EGFR-binding paratopic residues based on the reported crystal structure of the monobody-EGFR complex (PDB ID: 3QWQ) [15] for site directed-mutagenesis (Appendix A). We individually mutated the selected paratopic residues to histidine to decrease affinity, introduced a T49I mutation to increase affinity [15], and/or combined the individual mutations to generate variants with multiple mutations, resulting in a panel of anti-EGFR monobody variants with varying affinities to the human EGFR extracellular domain (EGFR-ECD), ranging from the equilibrium dissociation constant (*K*_D_) of 0.11 to 139 nM, as determined by bio-layer interferometry (Table 1). The dissociation rate constants (*k*_off_) of the variants accounted for the differences in affinities between the variants.

The resulting anti-EGFR monobody affinity variants were C-terminally fused via a GGS linker with the B-cell epitope-removed LO10 PE24 toxin [13,14] with C-terminal ER retention sequence of KDEL, generating five ER-PE24 IT variants (Figure 1A). Instead of the original ER retention sequence of REDLK, the most common ER retention sequence of KDEL was employed because it increased the cytotoxicity of PE24 by more efficient retrograde transport of the toxin to ER [17]. All constructed ER-PE24 ITs were expressed in *E.coli* and purified from the soluble fraction of cell lysates with high purity (≥98%) as determined by sodium dodecyl sulfate-polyacrylamide gel electrophoresis (SDS-PAGE) (Figure 1B).

The binding affinity of the purified ER-PE24 IT variants for human EGFR-ECD antigen was almost similar to that of each parent monobody variant (Figure 1C and Table 1), implying that the introduction of the PE24 toxin does not affect the binding affinity of the monobody for EGFR. We named the five ER-PE24 ITs as ER/X-PE24, where X indicates the nanomolar affinity of the IT for the EGFR antigen (Table 1). In ELISA with human or murine EGFR-ECDs, the purified ER-PE24 ITs, but not PE24 alone, showed a concentration-dependent binding activity for both human and murine EGFR-ECDs, with comparable binding strengths (Figure 1D). This provided evidence of the cross-reactivity of the anti-EGFR monobody variants with both human and murine EGFR. The rank order of the binding strength of ER-PE24 ITs determined using ELISA was consistent with that obtained by Bio-layer Interferometry (Table 1).

### 2.2. ER-PE24 IT Binds to EGFR-Expressing Cells According to EGFR Affinity and Cellular EGFR Expression Levels

In this study, we employed A431 and HT29 cell lines, which have been reported to express high levels (~1.8–3 × 10^5^ copies/cell) and low levels (~1–3 × 10^4^ copies/cell) of EGFR, respectively [18,19]. Particularly, the EGFR expression levels of HT29 cells were similar to that of normal skin cells (human epidermal keratinocyte, ~3.8 × 10^4^ copies/cell) [19]. We examined the cell surface expression levels of EGFR on the A431 and HT29 cell lines using flow cytometry. The results revealed a 10-fold higher expression of EGFR on A431 cells than in HT29 cells based on their relative mean fluorescence intensity (MFI) (Figure 2A), and were consistent with previous results [18,19]. Human embryonic kidney 293 (HEK293) cells have also been reported to have similar basal expression levels of EGFR as normal cells [20]. In this study, the EGFR expression profiles of the noncancerous HEK293 cells were very similar to those of HT29 cells (Figure 2A). The SW620 cells were confirmed as EGFR-negative cells (Figure 2A) [19]. Based on the above results, we considered the A431 and HT29 cell lines as model cells for EGFR high-expressing tumor cells and EGFR low-expressing normal cells, respectively.

We profiled the cell surface binding of ER-PE24 affinity variants to the model cell lines and found that the variants bound to A431 and HT29 cells in proportion to their concentrations and the EGFR binding affinity, the binding levels of which were much higher on A431 cells (Figure 2B) compared to those on HT29 cells (Figure 2C). All the ITs did not bind to EGFR-negative SW620 cells (Figure 2D), demonstrating their binding specificity. Additionally, a multiantigen nonspecificity enzyme-linked immunosorbent assay (ELISA) using four antigens (double-stranded DNA, insulin, hemocyanin, and cardiolipin; [21]) revealed that none of the ITs bound to any of the four off-target antigens even at a concentration of 500 nM (Appendix A), indicating that each IT lacks off-target specificity.

### 2.3. In Vitro Therapeutic Window of ER-PE24 Varies Depending on the Affinity for EGFR

To test the cytotoxic selectivity of ER-PE24 ITs for cells based on their EGFR expression levels, we selected the A431, HT29, and SW620 cell lines. The cells were treated with ER-PE24 ITs for 72 h at various concentrations, and the cell viability was determined. All the ER-PE24 ITs exhibited substantial cytotoxic activity for the two EGFR-expressing cells in a dose-dependent manner. The cytotoxicity was in proportion to their affinity for EGFR, with IC_50_ (defined as 50% inhibitory concentration of cell viability) ranging from 0.03 nM to 0.56 nM for EGFR high-expressing A431 cells and from 0.2 nM to 19 nM for EGFR low-expressing HT29 cells (Figure 3A and Table 2). In contrast, ER-PE24 ITs were slightly cytotoxic for EGFR-negative SW620 cells, with a less than 20% reduction in cell viability observed at the highest concentration of 400 nM (Figure 3A). These results demonstrated the specificity of ER-PE24 ITs for EGFR-expressing cells in which the anti-EGFR monobody moiety mediates the cellular internalization of the PE24 payload through EGFR-mediated endocytosis to kill cells. Furthermore, the strong correlation between the anti-EGFR affinity and IC_50_ values of ER-PE24 (Table 2) suggests that the enhanced binding to EGFR based on the affinity is primarily linked to the internalization efficacy, which then leads to the increased cytotoxicity of ER-PE24.

To compare the therapeutic and toxic doses amongst the ITs, the therapeutic index of each IT was obtained by determining the ratio of the IC_50_ of A431 cells with high levels of EGFR to that of HT29 cells with low basal levels of EGFR, i.e., [IC_50_ against HT29 cells]/[IC_50_ against A431 cells] [19]. We also estimated the therapeutic window of each ER-PE24 using the IC_20_ values (defined as 20% inhibitory concentration of cell viability) for A431 and HT29 cells, which were considered as the minimum effective dose (MED) and maximum tolerated dose (MTD), respectively [11]. The therapeutic index of ER-PE24 ITs increased with an inverse relationship to the EGFR affinity until a peak value of *K*_D_ ≈ 21 nM was reached and then declined with considerably lowering affinity (Figure 3B and Table 2). Accordingly, ER/21-PE24, with an intermediate affinity (*K*_D_ ≈ 21 nM), exhibited the highest therapeutic index of 54, which was ~ eightfold and ~ 1.6-fold higher than that of ER/0.2-PE24 with the highest affinity (*K*_D_ ≈ 0.2 nM) and ER/104-PE24 with the lowest affinity (*K*_D_ ≈ 104 nM), respectively. ER/21-PE24 also exhibited the broadest therapeutic window of 73 pM–1.5 nM, compared to the narrowest therapeutic window (19 pM–93 pM) of ER/0.2-PE24 (Figure 3C and Table 2). These results demonstrated that ER/21-PE24 with an intermediate affinity has a wide therapeutic window through selective cytotoxic activity against EGFR high-expressing cells and is not toxic against EGFR low-expressing cells in vitro, while ER/0.2-PE24 with the highest subnanomolar affinity can kill both EGFR high- and EGFR low-expressing cells with a narrow selectivity.

### 2.4. In Vivo Antitumor Activity and Toxicity of ER/0.2-PE24 and ER/21-PE24 in a Xenograft Model

ER-PE24 ITs demonstrated cross-reactivity with both human and murine EGFR. Therefore, we simultaneously compared the antitumor activity and systemic toxicity between ER/0.2-PE24 and ER/21-PE24 in BALB/c nude mice bearing EGFR-overexpressing A431 xenografts. When the mean tumor volume reached approximately 100–120 mm^3^, the tumor-bearing mice (TBM) were randomly assigned to a treatment group and were given intravenously (i.v.) administered various doses of ER/0.2-PE24, ER/21-PE24, or PBS buffer (as vehicle control) every other day. The systemic toxicity was monitored by observing death and weight loss in TBM. Subsequently, blood, liver, and kidney samples were collected at treatment endpoints to measure serum levels of alanine aminotransferase (ALT) and aspartate aminotransferase (AST) and for histological analysis. The MTD of ER-PE24 was determined as the dose at which no toxicity symptoms were observed. A single dose of 1 mg/kg (mpk) of ER/0.2-PE24 or ER/21-PE24 caused the death of all TBM (*n* = 6 per group) within 1 or 2 days, reflecting the severe on-target/off-tumor toxicity of these ITs. Thus, we lowered the doses to 0.6, 0.4, 0.2, and 0.1 mpk. A dose of 0.6 or 0.4 mpk ER/0.2-PE24 resulted in death in more than 75% TBM when administered twice (on day 4), although an initial inhibition of tumor growth was observed (Figure 4A and Table 3). The lower doses (0.2 or 0.1 mpk) of ER/0.2-PE24 markedly slowed the tumor growth showing ~63% and ~54% tumor growth inhibition (TGI), respectively, at the end of treatment compared with the PBS-treated controls. However, ER/0.2-PE24 resulted in dose-limiting severe adverse effects, including mortality of 12.5% TBM (1 out of 8 TBMs) after the first dose of 0.2 mpk (Figure 4A and Table 3), and a substantial increase in serum AST levels but with no increase in ALT in the TBM treated with both 0.1 and 0.2 mpk (Figure 4C), indicating damages to many organs, such as the liver, heart, skeletal muscle, kidney, pancreas, and lungs [22,23]. A histopathological examination of liver and kidney tissues of TBM treated with 0.2 mpk ER/0.2-PE24 showed tissue damage characterized by marked hepatocyte ballooning, inflammatory cell infiltration in the liver, and interstitial hemorrhage, tubular necrosis, and inflammatory cell infiltration in the kidney (Figure 4D and Appendix A) [23,24]. In contrast, 0.1 mpk of ER/0.2-PE24 showed no such noticeable adverse effects compared to the PBS-treated control, except for slight elevations in serum AST levels. Based on the above results, the MTD for ER/0.2-PE24 in this regimen was estimated to be 0.1 mpk.

In case of ER/21-PE24, mortality was observed in more than 50% of TBM after two doses at 0.6 mpk (Figure 4B and Table 3). However, the lower doses of 0.4, 0.2, and 0.1 mpk substantially inhibited the growth of tumors, showing ~79%, ~69%, and ~57% TGI, respectively, compared with the PBS-treated controls (Figure 4B). Importantly, with the above dosing regimen ranging from 0.4 to 0.1 mpk, no mortality and noticeable adverse effects were observed in TBM. In addition, the lack of significant differences in serum levels of ALT and AST as well as the absence of abnormal histological features in the liver and kidneys of the TBM at the end of treatments as compared to those of the PBS-treated TBM were observed (Figure 4B–D, Table 3, and Appendix A). Although TBMs treated with 0.4 mpk ER/21-PE24 showed a mild body weight loss in the middle of the treatment, they regained weight by the end of treatment (Figure 4B), indicating a good tolerance. Accordingly, the MTD of ER/21-PE24 in this regimen was estimated to be 0.4 mpk, which was fourfold higher than that of ER/0.2-PE24. Taken together, the above results demonstrated that, given the limited dosing regimen of our study, ER/21-PE24 has at least a fourfold improved therapeutic index compared to ER/0.2-PE24.

## 3. Discussion

The expression of tumor antigen target receptors on normal cells/tissues, such as EGFR targeted in this study, causes on-target/off-tumor toxicity, which is a major limitation of ITs for cancer therapy [3]. To overcome this barrier, we screened ER-PE24 ITs with an optimal affinity to EGFR to selectively target EGFR-overexpressing tumor cells, while sparing normal tissue cells that express low basal levels of EGFR. Compared with ER/0.2-PE24, which had the highest affinity for EGFR, ER/21-PE24 with an intermediate affinity to EGFR exhibited maximum selectivity for EGFR high-expressing cells over EGFR low-expressing cells, and thus demonstrated a much greater therapeutic window with high potency in both in vitro and in vivo models. The use of anticancer drugs with a narrow therapeutic window can be challenging in clinical practice, and their plasma concentrations often must be monitored to avoid toxic adverse effects. The findings of our study not only support the future clinical development of ER/21-PE24 but also offer a potentially widely applicable strategy for ITs targeting solid tumor receptors which are limited by on-target/off-tumor adverse effects.

Previous approaches have used antibodies or ligands with the highest affinity to a cancer antigen target receptor for IT development, including EGFR-targeted ITs [1,2]. Unfortunately, many normal tissues also express EGFR at low levels, resulting in toxicity from EGFR-targeted therapeutics, including antibodies, antibody-drug conjugates (ADCs), and ITs [3]. EGFR-targeting ITs have more restrictive criteria for target selection compared to ADC due to the cytotoxic mechanism of ITs that also kills non-dividing quiescent normal cells [25]. In this regard, many EGFR targeting ITs, based on EGFR ligand or high-affinity anti-EGFR scFv targeting wild-type EGFR, exhibited dose-limiting adverse effects [7,8], and most of them failed clinical trials for solid tumors therapy [2]. One approach to avoid EGFR-associated on-target/off-tumor toxicity is to target the EGFR variant III (EGFRvIII), which is selectively expressed in cancers, such as malignant glioblastoma [26,27,28]. In this study, we modulated the affinity of the EGFR-targeting moiety of ER-PE24 ITs to distinguish EGFR high-expressing cells from EGFR low-expressing cells, in an attempt to mitigate its adverse effects on normal cells. ER/0.2-PE24, which has the highest affinity for EGFR and a slow dissociation rate (*k*_off_), can bind to EGFR-expressing cells even when their expression levels are as low as in normal cells, potentially causing toxicity. In contrast, ER/21-PE24, which has an intermediate affinity for EGFR and a relatively fast dissociation rate, tends to detach from normal cells with low EGFR density, but remains on tumor cells with high EGFR density, enabling it to distinguish tumor cells from normal cells. This inverse relationship between EGFR affinity and cytotoxicity to high and low EGFR-expressing cells was observed in our in vitro cell cytotoxicity assays. A similar approach was employed to develop an IT of Her2/neu scFv-gelonin toxin [29] and an EGFR-targeting ADC of RN765C [19] to mitigate on-target/off-tumor toxicity by modulating the affinity of the cancer antigen targeting moiety. Aside from adjusting the tumor-antigen affinity of the ITs to the density of target antigens, other strategies, such as target cell-restricted reconstituting split toxin-embedded ITs [30] and tumor microenvironment-conditioned activating ITs [3], can increase the therapeutic window by reducing systemic toxicity.

The ITs currently in use have been based mainly on the targeting ligand and antibody specific to a cancer target in humans, and do not cross-react with their murine counterparts, causing challenges in determining the systemic toxicity of ITs in mice. In our study, we used an anti-EGFR monobody moiety with cross-reactivity to both human and murine EGFR to enable the simultaneous assessment of antitumor efficacy and adverse effects of ER-PE24 ITs in BALB/c nude mice bearing the EGFR-overexpressing A431 xenograft. We performed a dose-escalation study by i.v. injection of ER/0.2-PE24 and ER/21-PE24 into the TBM at doses ranging from 0.6 to 0.1 mpk every other day. ER/0.2-PE24 with the highest EGFR affinity exhibited substantial toxicities such as TBM death, body weight loss, and damage to liver and kidney tissues at doses greater than or equivalent to 0.2 mpk and was tolerated at a dose of 0.1 mpk. In contrast, ER/21-PE24 with intermediate EGFR affinity showed significant toxicity only at a dose of 0.6 mpk and elicited potent antitumor activity at 0.4 mpk and the lower doses without noticeable adverse effects. Thus, ER/21-PE24 showed a fourfold increase in MTD compared to ER/0.2-PE24. Our results suggest that ER/0.2-PE24, with a high EGFR affinity, can distribute in normal cells and tissues, resulting in significantly increased systemic toxicity compared to ER/21-PE24 with intermediate EGFR affinity.

At doses higher than MTD, both ER/0.2-PE24 and ER/21-PE24 caused TBM death before significant body weight loss, indicating their acute renal toxicity [24]. ER-PE24 ITs as small as ~35 kDa appeared to undergo rapid renal filtration, resulting in fatal kidney damage, as observed with PE38-carrying SS1P IT [24]. To minimize acute renal toxicity by reducing renal filtration, ER-PE24 IT needs to be engineered further to increase serum half-life, for example by conjugation of polyethylene glycol [31] or reformatting into an Fc-containing IgG. Even in the case of ITs with extended serum half-lives, the tumor antigen targeting affinity should be tailored depending on differences in the expression levels between tumors and normal tissues to avoid on-target/off-tumor toxicity and to enhance intratumoral penetration [32].

In conclusion, we compared the antitumor efficacy and toxicity of a panel of EGFR-targeting monobody-based PE24 toxins with different affinities to EGFR and found that ER/21-PE24 with intermediate EGFR affinity exhibits potent antitumor activity with little systemic toxicity and an ~eightfold and ~fourfold improved therapeutic index in vitro and in vivo, respectively, compared to ER/0.2-PE24 with the highest EGFR affinity. It is known that many solid tumors overexpress EGFR on the cell surface, but there is some heterogeneity between and within solid tumors in terms of EGFR expression levels [33]. Our results underscore the importance of adjusting the EGFR affinity of EGFR-targeted ITs based on EGFR expression levels on target tumors to maximize the therapeutic window, while reducing the on-target/off-tumor adverse effects.

## 4. Materials and Methods

### 4.1. Cell Lines

The human cell lines (A431, HT29, SW620) were purchased from Korea Cell Line Bank (Daejeon, Republic of Korea). The A431 cell line is a human epidermoid carcinoma cell line that highly expresses EGFR [18,19], whereas HT29 is a human colorectal carcinoma cell line that expresses lower levels of EGFR [18,19]. SW620 is a human colorectal adenocarcinoma cell line with a negative EGFR expression [19]. A431 was maintained in Dulbecco’s Modified Eagle’s Medium (DMEM; Cytiva, Uppsala, Sweden) and HT29 and SW620 were maintained in RPMI-1640 (Cytiva). All cell lines were cultured in a growth medium that was supplemented with 10% heat-inactivated fetal bovine serum (FBS; Cytiva), penicillin (100 U/mL; WelGene, Gyeongsan, Republic of Korea), streptomycin (100 μg/mL; WelGene), and amphotericin B (0.25 μg/mL; WelGene). All cell lines were authenticated by DNA short tandem repeat profiling (ABION CRO, Seoul, Republic of Korea) and used within 15 passages. All cell lines were maintained at 37 °C in a humified 5% CO_2_ incubator and were routinely screened for Mycoplasma contamination (CellSafe, Yongin, Republic of Korea).

### 4.2. Plasmid Construction

DNA encoding the B-cell epitope-removed 24-kDa PE fragment LO10 PE24 (residues 274–284 and residues 395–613) [13], with the C-terminal ER retention sequence ^609^REDLK^613^ substituted with ^609^KDEL^612^ and a DNA encoding the EGFR-targeting fibronectin scaffold monobody, adnectin 1 [15], was synthesized (Macrogen, Seoul, Republic of Korea). The DNAs encoding the modified PE24- and adnectin 1 were fused via a GGS linker by overlapping PCR and subcloned in-frame into the bacterial expression vector pET23 (Novagen) between *Nhe*I/*EcoR*I sites to have the N-terminal 6 × His tag for purification, resulting in ER-PE24 ITs. Similarly, the anti-EGFR monobody variants were subcloned in-frame into the pET23 to obtain an N-terminal 6 × His tag for purification. For the expression of human EGFR-ECD (residues 25–645)-fused Fc protein, EGFR-ECD-Fc, a cDNA (HG10001-UT) encoding human EGFR-ECD was purchased from Sino Biological (Beijing, China) and subcloned in-frame at *Not*I/*Hind*III sites of pcDNA3.4 to be expressed with C-terminal of the Fc region of immunoglobulin. All the constructs were confirmed by sequencing (CosmoGenetech, Seoul, Republic of Korea).

### 4.3. Protein Expression and Purification

All ER-PE24 ITs and anti-EGFR monobody affinity variants were solubly expressed at 20 °C for 48 h in *E. coli* strain SoluBL21(DE3) (Genlantis, San Diego, CA, USA) [15], purified using Ni-NTA resin (Qiagen, Hilden, Germany; 30210), and finally formulated in a phosphate-buffered saline (PBS) buffer (2.67 mM KCl, 1.47 mM KH_2_PO_4_, 137 mM NaCl, 8.1 mM Na_2_HPO_4_, pH 7.4) [34].

For the expression of EGFR-ECD-Fc protein, the corresponding plasmid was transiently transfected into HEK293F cell cultures in Freestyle 293F medium (Invitrogen, Waltham, MA, USA) according to the standard protocol [35]. Human EGFR-ECD-Fc protein was purified from the culture supernatants using protein-A agarose resin (Captiva PriMAB, Repligen, Waltham, MA, USA) and finally formulated in a PBS buffer. The concentration of purified proteins was determined using the Bicinchoninic Acid (BCA) assay.

### 4.4. ELISA

The binding specificity of ER-PE24 ITs to EGFR-ECD proteins and four off-target antigens (double-stranded DNA, insulin, hemocyanin, cardiolipin) were determined using ELISA. The 96-well EIA/RIA plates (Corning, New York, NY, USA) were coated for 1 h at 25 °C with 250 μg/well (5 μg/mL, 50 μL) of the recombinant human EGFR-ECD (Sino Biological, 10001-H08H), mouse EGFR-ECD (Sino Biological, 51091-M08H), double-stranded DNA (1 μg/mL, Sigma-Aldrich, St. Louis, MI, USA; D4522;), insulin (5 μg/mL, Sigma-Aldrich, I9278;), cardiolipin (50 μg/mL, Sigma-Aldrich, C0563;), or hemocyanin (5 μg/mL, Sigma-Aldrich, H8283;) and blocked using the blocking solution [PBST (PBS, 0.05% (*v/v*) Tween-20, pH 7.4), 3% (*w/v*) skim milk]. After washing thrice with PBST, ER-PE24 ITs were added to each well in fivefold serial dilutions from 400 nM to 1.4 pM for EGFR-ECD binding, or from 500 nM to 100 nM for the off-target antigen binding, for 1 h at 25 °C. After washing with PBST, bound ER-PE24 ITs were detected by rabbit anti-pseudomonas exotoxin A (Sigma-Aldrich, St. Louis, MI, USA; P2318; dilution 1:3000) followed by HRP conjugated-anti-rabbit IgG antibody (Cell Signaling Technology, Danvers, MA, USA; 7074; dilution 1:2000) and subsequently incubated with ultra TMB (3,3′,5,5′-tetramethylbenzidine)-ELISA solution (ThermoFisher Scientific, Waltham, MA, USA; 34028). Absorbance was read at 450 nm on a Multiskan GO microplate reader (ThermoFisher Scientific). The results are presented after the subtraction of the background control value.

### 4.5. Bio-Layer Interferometry

The kinetic interactions of human EGFR-ECD-Fc with ER-PE24 ITs were observed with the Octet QKe System (Sartorius, Göttingen, Germany), as previously described [21,36]. All kinetic experiments were performed at 30 °C with orbital shaking at 1000 rpm with 200 μL in black 96-well flat-bottom plates (Greiner Bio-One, Frickenhausen, Germany). All reagents were diluted with binding buffer (PBS, 0.02% (*v/v*) Tween-20, pH 7.4). Anti-human IgG Fc capture biosensors (Sartorius) were equilibrated for 10 min in a binding buffer. After equilibrium was reached, the biosensors were loaded with ER-PE24 protein (5 μg/mL) for 1 min. A 30-s baseline was then carried out in a binding buffer before the association for 3 min at a variety of indicated concentrations of ER-PE24. Dissociation was then monitored for the indicated time in the binding buffer. A blank reference sensor without ER-PE24 was adopted for all experiments to account for the non-specific binding of the analyte to the sensor. The association (*k*_on_) and dissociation rate (*k*_off_) constants as well as the equilibrium dissociation constant (*K*_D_) were determined by fitting to sensorgrams via the 1:1 binding model in Octet Data Analysis software version 11.0 (Sartorius).

### 4.6. Cell Surface EGFR Binding Assay

The cell surface expression of EGFR was determined by flow cytometry following the indirect immunofluorescent labeling of cells with 30 nM of mouse anti-EGFR Ab30 antibody (Abcam, Cambridge, UK) and then with Alexa Fluor 488-conjugated anti-mouse IgG antibody (Invitrogen, A11001, dilution 1:600). To determine the concentration-dependent binding of ER-PE24 ITs, the cells were incubated with each IT for 1 h at 4 °C and the bound ITs were then detected by rabbit anti-pseudomonas exotoxin A (Sigma-Aldrich, P2318, dilution 1:3000) followed by Alexa647-conjugated anti-rabbit IgG antibody (Invitrogen, A21245, dilution 1:600). After washing with PBS supplemented with 1% BSA, the cells were analyzed on the FACSCalibur flow cytometer (Becton Dickinson, Franklin Lakes, NJ, USA). Flow cytometric data were analyzed using the FlowJo v10 software (Tree Star) to calculate MFI [35].

### 4.7. Cell Viability Assay

A431, HT29, and SW620 cells were seeded at a density of 5.0 × 10^3^ cells per well in 96-well culture plates (SPL Life Sciences, 30096) for 24 h and then treated for 72 h with various concentrations of the indicated ER-PE24 resuspended in complete medium. The cell viability was measured using a CellTiter-Glo luminescent cell viability assay (Promega, Madison, WI, USA) according to the manufacturer’s instructions [21]. The luminescence was measured with the Cytation 3 multimode reader (BioTek, Winooski, VT, USA). Readings were normalized to that of the PBS buffer control. The cell viability percentage was calculated based on complete (100%) cell killing that was achieved using 2.3 μg/mL of staurosporine (Sigma-Aldrich) as a control. The IC_50_ and IC_20_ values (the concentration inhibiting cell viability by 50% and 20%, respectively) were evaluated by the four-parameter sigmoidal equation using the GraphPad Prism 8 software [19,21].

The therapeutic index of each ER-PE24 was calculated using the IC_50_ values for A431 and HT29, which represent EGFR-overexpressing target tumor cells and low EGFR-expressing normal cells, respectively, using the following formula [19]: Therapeutic index = [IC_50_ against HT29 cells]/[IC_50_ against A431 cells].

The therapeutic window of each variant of ER-PE24 was determined by the range between the IC_20_ values for A431 and HT29, which were considered as minimum effective dose (MED) and maximum tolerated dose (MTD), respectively [11]: Therapeutic window = [IC_20_ against A431 cells]–[IC_20_ against HT29 cells].

### 4.8. Mouse Xenograft Tumor Model

All animal experiments were approved by Ajou University’s Animal and Ethics Review Committee (approval ID: 2022-0008) and were conducted in accordance with the guidelines of the Animal Care and Use Committee of Ajou University. For tumor xenograft models, 4–6-week-old female BALB/c nude mice (Orient Bio, Seongnam, Republic of Korea) weighing 16– 18 g were subcutaneously injected with A431 cells (5 × 10^6^ cells per mouse) in 100 μL of a 1:1 mixture of PBS and Matrigel (Corning, 354234). Once the mean tumor volume reached approximately 100–120 mm^3^, the mice were randomly assigned to treatment groups and an ER-PE24 IT or vehicle control (PBS), as indicated in the Figure legend, was administered via the tail vein in a dose/weight-adjusted manner. Tumor volume (*V*) was calculated by the formula *V* = *L* × *W^2^*/2, where *L* and *W* are the long and short lengths of the tumor, respectively [21,35]. The animals were withdrawn from the study when the tumor volume was 2000 mm^3^. The TGI rate of ER-PE24 compared to that of the PBS control was determined on the last day of the study using the formula TGI (%) = [100 − (V_f_^ER-PE24^ − V_i_^ER-PE24^)/(V_f_^PBS^–V_i_^PBS^) × 100], where V_i_ is the initial mean tumor volume in the ER-PE24 or PBS treatment group and V_f_ is the final mean tumor volume in the ER-PE24 or PBS treatment group [21,35].

On day 2 after the last treatment, the mice were euthanized by CO_2_ asphyxiation, and the livers, kidneys, and blood samples were collected. Livers and kidneys were fixed in 10% formalin for 24 h. They were subsequently sectioned and stained with hematoxylin and eosin (Ajou University Institute of Medical Science) [21]. The levels of ALT and AST in blood serum were measured using transaminase kits (AM102-K for ALT and AM103-K for AST; Asan Pharmaceutical Co., Seoul, Republic of Korea) according to the manufacturer’s instructions [37].

### 4.9. Statistical Analysis

Data for imaging experiments are presented using representative images. The mean ± SEM for pooled data or the mean ± SD of at least triplicate experiments, unless specified otherwise, are presented. The one-way analysis of variance (ANOVA) with Holm-Sidak multiple-comparisons post-hoc test was performed using GraphPad Prism to compare the test and control groups of the in vivo tumor growth experiments. A *p*-value of less than 0.05 was considered statistically significant.

## Figures and Tables

**Figure 1 ijms-23-15820-f001:**
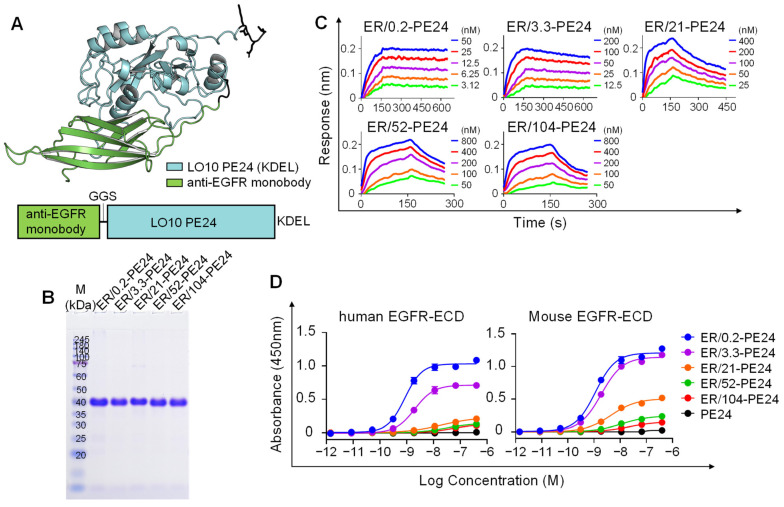
Generation and characterization of ER-PE24 IT variants with different affinities for EGFR. (**A**) Schematic view of the ER-PE24 IT construction (lower panel), and predicted structure via AlphaFold (AlphaFold v.2.0) (upper panel). The anti-EGFR monobody (Green) was fused to the N-terminus of the B-cell epitope-removed PE24 toxin, LO10 PE24 (Cyan), via a GGS linker. (**B**) SDS-PAGE analyses of the purified ER-PE24 ITs (each 5 μg loading) under non-reducing conditions. The gel was stained with Coomassie Blue R-250. (**C**) Representative binding isotherms of the immobilized EGFR-ECD-Fc in relation to soluble ER-PE24 IT, as measured by bio-layer interferometry. The concentrations of ER-PE24 IT analyzed are indicated using different colors. The binding kinetics parameters are listed in Table 1. (**D**) Concentration-dependent binding activities of ER-PE24 ITs to human or murine EGFR-ECD, as determined by ELISA. Error bars represent the mean ± SEM (*n* = 3).

**Figure 2 ijms-23-15820-f002:**
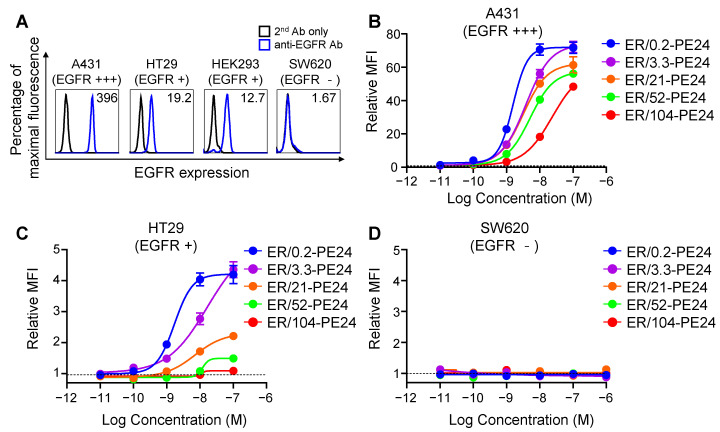
Cell surface binding activities of ER-PE24 ITs based on the affinity for EGFR and EGFR expression levels on cells. (**A**) Representative histograms showing EGFR expression levels on the surface of indicated cell lines determined using flow cytometry. Negative control was used only with the secondary antibody (2nd Ab only) without the anti-EGFR primary antibody. The relative mean fluorescence intensity (MFI) was calculated by comparison with the MFI of negative control and is presented in each histogram. (**B–D**) Concentration-dependent cell surface binding activities of ER-PE24 ITs to three different cell lines, EGFR high-expressing A431 cells (**B**), EGFR low-expressing HT29 cells (**C**), and EGFR negative SW620 cells (**D**), determined by flow cytometry. The relative MFI was calculated by comparison with the MFI measured with or without ER-PE24 (indicated by the dashed black line). In (**B**–**D**), each value represents the mean ± SEM (*n* = 3).

**Figure 3 ijms-23-15820-f003:**
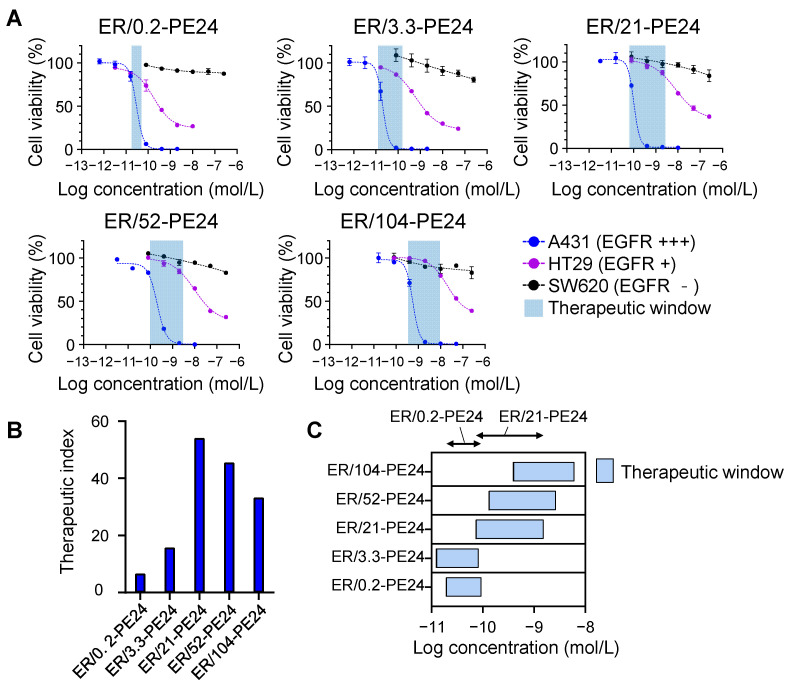
ER-PE24 ITs have different therapeutic windows in vitro depending on the anti-EGFR affinity. (**A**) In vitro cell cytotoxicity assay of ER-PE24 variants in EGFR high-expressing A431 cells, EGFR low-expressing HT29 cells, and EGFR negative SW620 cells. Each cell line was cultured with the indicated ER-PE24 ITs at various concentrations for 72 h, before the determination of cell viability using the CellTiter-Glo assay. Error bars represent the mean ± SD (*n* = 3). (**B**) The therapeutic index of each ER-PE24 was calculated by dividing [IC_50_ against HT29 cells] by [IC_50_ against A431 cells]. (**C**) The therapeutic window of each ER-PE24 was estimated by the range between the IC_20_ values against A431 and HT29, i.e., [IC_20_ against A431 cells]–[IC_20_ against HT29 cells]. The values of the therapeutic index and therapeutic window of each ER-PE24 variant are listed in Table 2.

**Figure 4 ijms-23-15820-f004:**
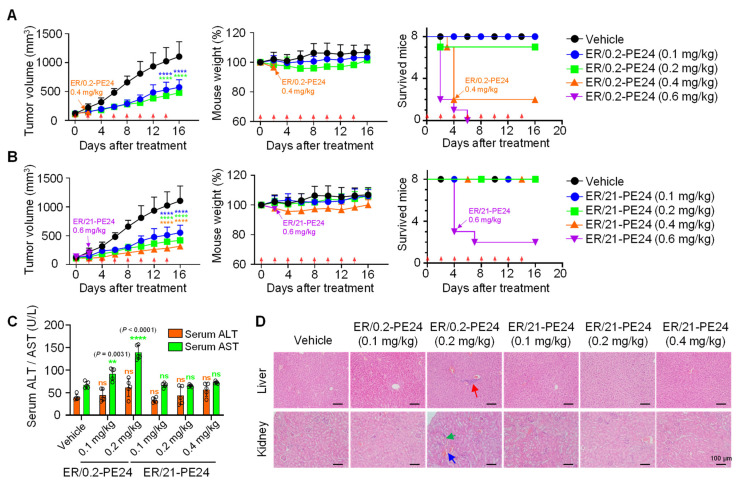
ER/21-PE24 with an intermediate EGFR affinity has a much broader therapeutic window than ER/0.2-PE24 with the highest EGFR affinity. (**A**,**B**) Tumor growth (**left**), mouse body weight (**middle**), and survival curves (**right**), as measured during treatment of A431 tumor-bearing mice (TBM) initiated at the tumor volume of ~100 to 120 mm^3^ with either i.v. injection of ER/0.2-PE24 (**A**) or ER/21-PE24 (**B**) at a dose of 0.1, 0.2, 0.4, and 0.6 mg/kg every other day (indicated by red arrows). The vehicle indicates PBS buffer as control. If TBM death occurs in more than 2 TBMs from each regimen, treatment was stopped, as indicated by the arrow in the panel. Symbols and error bars represent the mean ± SEM. Data were pooled from two independent experiments with four mice per group. (**C**) Serum levels of liver enzymes (ALT and AST) were measured in each group of TBM at the endpoint of treatment. Each symbol represents the value obtained from individual mice, and the bar graphs present the mean ± SD (*n* = 5 per group). In (**A**–**C**), **, *p* < 0.01; ****, *p* < 0.0001 versus the vehicle-treated TBM. ns, not significant. (**D**) Hematoxylin and eosin (H&E) staining of representative tissue sections from the liver and kidney of each group of TBM was performed at the end of treatment. Abnormal histological features suggestive of hepatic toxicity or renal toxicity are indicated using arrows of different colors [Inflammatory cell infiltration (red), tubular necrosis (green), interstitial hemorrhage (blue)]. Magnification, ×200; scale bar, 100 μm.

**Table 1 ijms-23-15820-t001:** Binding kinetics of the interactions between either soluble anti-EGFR monobody variants or PE24-fused ER-PE24 IT variants and immobilized human EGFR-ECD, as measured by bio-layer interferometry.

Variants	Mutations in Anti-EGFR Monobody	Binding Kinetic Parameters ^(a)^		
		*K*_D_ (nM)	*k*_on_ (M^−1^s^−1^, ×10^4^)	*k*_off_ (s^−1^, ×10^−4^)
ER/0.2 ^(b)^	T49I	0.11 ± 0.03	14.0 ± 0.2	0.16 ± 0.04
ER/0.2-PE24		0.24 ± 0.05	24.3 ± 0.9	0.59 ± 0.12
ER/3.3 ^(b)^	Wild type	3.62 ± 0.08	7.44 ± 0.08	2.69 ± 0.05
ER/3.3-PE24		3.27 ± 0.14	9.82 ± 0.24	3.21 ± 0.12
ER/21 ^(b)^	Y29H, T49I, V54H, Y83H	41.5 ± 0.6	6.93 ± 0.09	28.7 ± 0.1
ER/21-PE24		21.3 ± 0.7	12.0 ± 0.4	25.5 ± 0.4
ER/52 ^(b)^	T49H, K79H	59.5 ± 1.2	5.43 ± 0.11	32.3 ± 0.2
ER/52-PE24		52.0 ± 2.6	7.57 ± 0.28	39.3 ± 1.28
ER/104 ^(b)^	Q30H	139 ± 3	7.71 ± 0.14	107 ± 1
ER/104-PE24		104 ± 4	6.41 ± 0.24	66.7 ± 1.4

^(a)^ Each value represents the mean ± SEM of two independent experiments. At least five data sets were used in each experiment to determine the binding kinetic constants. ^(b)^ Anti-EGFR monobody form without PE24 fusion.

**Table 2 ijms-23-15820-t002:** The IC_50_, therapeutic index, and therapeutic window of ER-PE24 ITs in vitro. ^(a)^

ITs	IC_50_ ^(b)^		Therapeutic Index ^(c)^	Therapeutic Window (nM) ^(c)^
	A431 (nM)	HT29 (nM)		
ER/0.2-PE24	0.0305 ± 0.0019	0.202 ± 0.020	6.6	(0.0192 ± 0.0022)–(0.0928 ± 0.0016)
ER/3.3-PE24	0.0186 ± 0.0018	0.294 ± 0.088	15.8	(0.0123 ± 0.0005)–(0.0823 ± 0.0320)
ER/21-PE24	0.112 ± 0.015	6.06 ± 1.48	54.1	(0.0732 ± 0.0060)–(1.52 ± 0.14)
ER/52-PE24	0.296 ± 0.005	10.3 ± 1.1	45.5	(0.130 ± 0.018)–(2.65 ± 0.08)
ER/104-PE24	0.562 ± 0.001	18.7 ± 2.4	33.3	(0.389 ± 0.064)–(6.07 ± 2.88)

^(a)^ The values of IC_50_ and therapeutic window are represented as the mean ± SEM of three independent experiments. ^(b)^ IC_50_ and IC_20_ are the 50% and 20% inhibitory concentrations of cell viability, respectively. ^(c)^ The therapeutic index of each ER-PE24 was calculated by dividing [IC_50_ against HT29 cells] by [IC_50_ against A431 cells]. The therapeutic window of each ER-PE24 was estimated from the IC_20_ values for A431 and HT29, i.e., [IC_20_ against A431 cells]–[IC_20_ against HT29 cells]. Each value represents the mean ± SEM of three independent experiments.

**Table 3 ijms-23-15820-t003:** In vivo toxicity of ER-PE24 ITs measured in A431 tumor-bearing mice.

ITs	Dose	Weight Loss (%)	Death/Total Mice
ER/0.2-PE24	0.6 mg/kg × 2	-	8/8
	0.4 mg/kg × 2	3.4	6/8
	0.2 mg/kg × 8	0	1/8
	0.1 mg/kg × 8	0	0/8
ER/21-PE24	0.6 mg/kg × 2	2.2	6/8
	0.4 mg/kg × 8	0	0/8
	0.2 mg/kg × 8	0	0/8
	0.1 mg/kg × 8	0	0/8

The body weight of the mice and the number of dead mice were measured at the end of the indicated treatment of A431 tumor-bearing mice (TBM) with either i.v. injection of ER/0.2-PE24 or ER/21-PE24 at a dose of 0.1, 0.2, 0.4, and 0.6 mg/kg every other day for a total of 2 doses (×2) or 8 doses (×8) (*n* = 8 per group), as shown in Figure 4A,B.

## Data Availability

All data in this study are available within the article or from the corresponding author upon reasonable request.

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
