# Peer review of "Expanding the Therapeutic Window of EGFR-Targeted PE24 Immunotoxin for EGFR-Overexpressing Cancers by Tailoring the EGFR Binding Affinity"

_ijms, 2022, doi:10.3390/ijms232415820_

Round 1

Reviewer 1 Report

The authors report a strategy to expand the therapeutic window of EGFR targeting immunotoxins for EGFR-overexpressing cancers by tailoring the EGFR binding affinity of the antibody fragment. Their data clearly show that the use of an intermediate affinity binder expands the therapeutic window compared to a high affinity binder both in vitro against cell line models and in vivo in mice. The data are convincing and the paper can be improved as follows. 

1)    Please eliminate redundancies concerning findings from the discussion. 

2)    Discussion should also address the following. Animals are dying without weight loss suggesting acute toxicity. Data could be interpreted to mean that renal toxicity is a major factor. Could it be that PE24 IT are rather small and being filtered into the kidney. Perhaps the use of a larger PE such as PE40 would present an advantage. How does this fit in with their premise that immunotoxins with intermediate affinities are superior?

3)    Despite the fact that immunotoxins with intermediate affinities have a greater therapeutic window, these windows are still rather narrow. How could this be addressed in the future? Different toxins? 

4)    In the discussion, the authors should state what they think the clinical future of this approach might be. 

Author Response

Please see the uploaded PDF file of the responses to your comments, which is prepared in color font for your convenience.

----------------------------------------------------

Responses to specific comments of reviewer #1

The authors report a strategy to expand the therapeutic window of EGFR targeting immunotoxins for EGFR-overexpressing cancers by tailoring the EGFR binding affinity of the antibody fragment. Their data clearly show that the use of an intermediate affinity binder expands the therapeutic window compared to a high affinity binder both in vitro against cell line models and in vivo in mice. The data are convincing and the paper can be improved as follows.

Comment#1-1) Please eliminate redundancies concerning findings from the discussion.

Response) Thank you for your comment. In response to your comment, we have deleted the following sentence in [Discussion] (p. 10, line 346) in the revised manuscript as follows:

While ER/0.2-PE24 with EGFR affinity of KD ≈ 0.2 nM showed the narrowest therapeutic window, ER/21-PE24 with EGFR affinity of KD ≈ 21 nM showed a ~8-fold greater therapeutic index compared with ER/0.2-PE24.

Comment#1-2) Discussion should also address the following. Animals are dying without weight loss suggesting acute toxicity. Data could be interpreted to mean that renal toxicity is a major factor. Could it be that PE24 IT are rather small and being filtered into the kidney. Perhaps the use of a larger PE such as PE40 would present an advantage. How does this fit in with their premise that immunotoxins with intermediate affinities are superior?

Response) We appreciate your comment. As you pointed out, PE toxin-carrying antibody fragment-based ITs have a molecular weight of ~30 – 60 kDa, which is smaller than that of the kidney cut-off (~60-66 kDa). Thus, they have a short serum half-life mainly due to rapid clearance by the kidney [Lei et al (2020) Proc Natl Acad Sci U S A 117:6086-6091]. In the kidney, the IT passes through the glomerulus and is taken up by the proximal tubular cells, thereby causing renal toxicity by damaging the proximal tubular cells [Lei et al (2020) Proc Natl Acad Sci U S A 117:6086-6091]. Though PE38-fused IT showed a little bit longer serum half-life (~19 min) than that (~13 min) of PE24-fused IT in mouse models (Weldon et al (2012) Mol Cancer Ther. 12(1): 48-57), it still has relatively short serum half-life. However, increasing the size of PE24-based IT by PEGylation substantially prolonged serum half-life and reduced its renal filtration [Zheng et al (2020) Mol Cancer Ther 19:812-821]. Even in the case of ITs with extended serum half-lives, tumor antigen targeting affinity should be adjusted depending on the expression levels on tumors and normal tissues to avoid on-target/off-tumor toxicity and enhance intratumoral penetration.

In response to your comment, we added a following paragraph in [Discussion] (p. 12) of the revised manuscript as follows:

“At doses higher than MTD, both ER/0.2-PE24 and ER/21-PE24 caused TBM death before significant body weight loss, indicating their acute renal toxicity [24]. ER-PE24 ITs as small as ~35 kDa appeared to undergo rapid renal filtration, resulting in fatal kidney damage, as observed with PE38-carrying SS1P IT [24]. To minimize acute renal toxicity by reducing renal filtration, ER-PE24 IT needs to be further engineered to increase serum half-life, for example by conjugation of polyethylene glycol [30] or reformatting into an Fc-containing IgG. Even in the case of ITs with extended serum half-lives, the tumor antigen targeting affinity should be tailored depending on differences in the expression levels between tumors and normal tissues to avoid on-target/off-tumor toxicity and to enhance intratumoral penetration [31].”

Comment#1-3) Despite the fact that immunotoxins with intermediate affinities have a greater therapeutic window, these windows are still rather narrow. How could this be addressed in the future? Different toxins?

Response) The narrow therapeutic index is the major hurdle in the development of ITs. In addition to tailoring the tumor antigen affinity of IT depending on target antigen density, other strategies such as tumor microenvironment-conditioned activating ITs and target cell-restricted reconstituting split toxin-embedded ITs (Purde et al. (2020) Proc Natl Acad Sci USA 117:22090–100; Antibody Therapeutics, 2022, 5:164–176) can increase the therapeutic window by reducing systemic toxicity.

In response to your comment, we added the following sentence in the [Discussion] (p. 11) of the revised manuscript:

“Aside from adjusting the tumor-antigen affinity of the ITs to the density of target antigens, other strategies, such as target cell-restricted reconstituting split toxin-embedded ITs [31] and tumor microenvironment-conditioned activating ITs [3], can increase the therapeutic window by reducing systemic toxicity.”

Comment#1-4) In the discussion, the authors should state what they think the clinical future of this approach might be.

Response) The expression levels of EGFR vary a lot among solid tumors [Kries (2019) Sci Rep 9:13564; Douillard et al. (2014) J Thorac Oncol 9:717-724], indicating the need to adjust the EGFR affinity of EGFR-targeted ITs based on EGFR expression levels on targeted tumors to maximize the therapeutic window while minimizing the on-target/off-tumor adverse effects.

In response to your comment, we edited the following sentence in [Discussion] (p. 12) of the revised manuscript as follows:

“It is known that many solid tumors overexpress EGFR on the cell surface, but there is some heterogeneity between and within solid tumors in terms of EGFR expression levels [33]. Our results underscore the importance of adjusting the EGFR affinity of EGFR-targeted ITs based on EGFR expression levels on target tumors to maximize the therapeutic window while minimizing the on-target/off-tumor adverse effects.

Reviewer 2 Report

The manuscript by Jun et al. describes the characterization of an immunotoxin (IT) targeted to the EGF receptor by a fibronectin domain. The catalytic activity of the IT is derived fom Pseudomonas exotoxin A.

Insetad of enhancing the affinity for the chosen receptor the authors tried to reduce the affinity which is so to say unusual. By modulating the affinity of the IT due to point mutations the authors want to enhance the safety of the IT. They define the therapeutic index of the respective IT variants by studying the activity on cells which express a low amount of EGFR (monitoring safty) and cells which highly express EGFR (monitoring activity). Using mouse experiments authors want to support their findings in vivo.

The approach is interesting and well described.

However, there are some points which have to be addressed prior publication:

1. Introduction/abstract:

Plese define the 24 kDa fragment (PE24). Which amino acids are chosen? For non-toxicologists, please explain the molecular mechanism of the toxin, why does it kill cells?

2. The definition of therapeutic index in pharmacology is LD5/ED95. Why did you chose IC20 and 50 values?

3. If you lower affinity towards the EGFR: Are there new off targets? You may show this by binding to A431 of HT29 knockout cells. The use of SW620 cells is not sufficient, because a possible binding partner (not present on SW620 cells) may be overlooked. Alternatively you may show whether the spectrum of proteins binding to the respective variants (from lysates) is similar or different.

4. Figure 3: The viability of HT29 in most experiments does not reach 100%. If the hioghest value would have been set to 100%, this should have major impact on the IC20 values. It is well known that expression of spare receptors enhance the sensitivity of tissues for ligands and dose response-curves shift to lower concentrations. However the curves of HT29 cells are compressed. Please comment on this.

5. Fig. 4: Due to the higher affinity of ER/0.2-PE24 lower IT concentrations should have been used. 0.1 mg/kg seems to be effective AND safe. Since the affinity is 20 times higher compared to ER/21-PE24 mouse experiments should have been performed with 10 to 20 times lower concentrations. The concentration range used is not optimal to draw the conclusion that the therapeutic window of the first IT is smaller compared to the second. The experiment only shows higher toxicity of the one with higher affinity what should be expected.

Author Response

Please see the uploaded PDF file of the responses to your comments, which contains the revised Figures and is prepared in color font for your convenience.

------------------------------------------

Responses to specific comments of reviewer #2

The manuscript by Jun et al. describes the characterization of an immunotoxin (IT) targeted to the EGF receptor by a fibronectin domain. The catalytic activity of the IT is derived from Pseudomonas exotoxin A.

Instead of enhancing the affinity for the chosen receptor the authors tried to reduce the affinity which is so to say unusual. By modulating the affinity of the IT due to point mutations the authors want to enhance the safety of the IT. They define the therapeutic index of the respective IT variants by studying the activity on cells which express a low amount of EGFR (monitoring safty) and cells which highly express EGFR (monitoring activity). Using mouse experiments authors want to support their findings in vivo.

The approach is interesting and well described.

However, there are some points which have to be addressed prior publication:

Comment#2-1) Introduction/abstract: Please define the 24 kDa fragment (PE24). Which amino acids are chosen? For non-toxicologists, please explain the molecular mechanism of the toxin, why does it kill cells?

Response) Thank you for your comment. In response to your comment, we added the following sentences in [Introduction] (p. 2) of the revised manuscript as follows:

“As a model toxin, we used a de-immunized 24-kDa PE fragment variant, LO10 PE24 [13, 14], which contains the furin cleavage site in domain II (residues 274-284) and the domain III (395-613) of PE with a replacement of the C-terminal endoplasmic reticulum (ER) retention sequence 609REDLK613 with 609KDEL612. When PE24 is internalized into endosomes by its targeting moiety, it escapes into the cytosol through the retrograde transport system and inactivates eukaryotic elongation factor 2 by catalyzing adenosine diphosphate (ADP) ribosylation, causing inhibition of protein synthesis and resulting in cell death [13, 14].”

Comment#2-2) The definition of therapeutic index in pharmacology is LD5/ED95. Why did you chose IC20 and 50 values?

Response) Therapeutic index is generally defined as the ratio between the therapeutic dose and the toxic dose of a given drug and is generally calculated by dividing the therapeutic dose for ED50 by the toxic dose for TD50 or LD50 [McCallum L et al (2014) Handbook of pharmacogenomics and stratified medicine. London: Academic Press; 365-83]. LD10/ED90 or more restrictively LD5/ED95 is defined as the safety margin or safety index [Chou (2006) Pharmacol Rev 58: 621-681 / Zhou et al (2013) Br J Anaesth 111: 825-832]. In our study, the therapeutic index of each IT was determined by TD50/ED50, where TD50 was IC50 against HT29 cells and ED50 was IC50 against A431 cells, i.e., [IC50 against HT29 cells] / [IC50 against A431 cells]. For the therapeutic window, we used the IC20 values, i.e., [IC20 against A431 cells] – [IC20 against HT29 cells], following a previous report [Fernandes et al (2020) Nat Commun 11, 3157].

The above definition was described in the original manuscript's Materials and Methods section.

Comment#2-3) If you lower affinity towards the EGFR: Are there new off targets? You may show this by binding to A431 of HT29 knockout cells. The use of SW620 cells is not sufficient, because a possible binding partner (not present on SW620 cells) may be overlooked. Alternatively you may show whether the spectrum of proteins binding to the respective variants (from lysates) is similar or different.

Response) We agree with your comment. Unfortunately, EGFR-knockout A431 and HT29 cells are not available. To address your comment, we have determined the non-specific binding of each IT variant to four off-target antigens [double-stranded DNA, insulin, keyhole limpet hemocyanin, and cardiolipin] by enzyme-linked immunosorbent assay (ELISA) followed by previously published methods [Shin et al (2020) Science Advance 6: eaay2174]. As you can see below, none of the ITs bound to any of the four off-target antigens even at a concentration of 500 nM, indicating that each IT lacks off-target specificity.

                The above data was added in the [Supplementary Figure 2] and described in [Result] and [Materials and Method] in the revised manuscript,

 “Supplementary Figure 2. Evaluation of nonspecific binding of the indicated ER-PE24 ITs to four off-target antigens (EGFR-ECD, double-stranded DNA (dsDNA), insulin, hemocyanin, and cardiolipin), as determined by ELISA. EGFR-ECD was used as a positive control. Error bars represent the mean ± SD.”

In [Result] in p. 5 (lines 168-172) of revised manuscript,

“Additionally, a multiantigen nonspecificity enzyme-linked immunosorbent assay (ELISA) using four antigens (double-stranded DNA, insulin, hemocyanin, and cardiolipin; [21]) revealed that none of the ITs bound to any of the four off-target antigens even at a concentration of 500 nM (Supplementary Figure 2), indicating that each IT lacks off-target specificity.”

In [Materials and Methods] in p. 13-14 (lines 433-451) of revised manuscript,

“ELISA

The binding specificity of ER-PE24 ITs to EGFR-ECD proteins and four off-target antigens (double-stranded DNA, insulin, hemocyanin, cardiolipin) were determined using ELISA. The 96-well EIA/RIA plates (Corning) were coated for 1 h at 25 °C with 250 μg/well (5 μg/mL, 50 μL) of the recombinant human EGFR-ECD (Sino Biological, 10001-H08H), or mouse EGFR-ECD (Sino Biological, 51091-M08H), double-stranded DNA (1 μg/mL, Sigma-Aldrich, St.Louis, MI, USA; D4522;), insulin (5 μg/mL, Sigma-Aldrich, I9278;), cardiolipin (50 μg/mL, Sigma-Aldrich, C0563;), or hemocyanin (5 μg/mL, Sigma-Aldrich, H8283;) and blocked using the blocking solution [PBST (PBS, 0.05 %(v/v) Tween-20, pH 7.4), 3 %(w/v) skim milk]. After washing thrice with PBST, ER-PE24 ITs were added to each well in 5-fold serial dilutions from 400 nM to 1.4 pM for EGFR-ECD binding, or from 500 nM to 100 nM for the off-target antigen binding, for 1 h at 25 °C. After washing with PBST, bound ER-PE24 ITs were detected by rabbit anti-pseudomonas exotoxin A (Sigma-Aldrich, P2318; dilution 1:3000) followed by HRP conjugated-anti-rabbit IgG antibody (Cell Signaling Technology, Danvers, MA, USA; 7074; dilution 1:2000) and subsequently incubated with ultra TMB (3,3’,5,5’-tetramethylbenzidine)-ELISA solution (ThermoFisher Scientific, Waltham, MA, USA; 34028). Absorbance was read at 450 nm on a Multiskan GO microplate reader (ThermoFisher Scientific). The results are presented after subtraction of the background control value.”

Comment#2-4) Figure 3: The viability of HT29 in most experiments does not reach 100%. If the highest value would have been set to 100%, this should have major impact on the IC20 values. It is well known that expression of spare receptors enhance the sensitivity of tissues for ligands and dose response-curves shift to lower concentrations. However the curves of HT29 cells are compressed. Please comment on this.

Response) Thank you for pointing it out. To address your comment, we repeated the experiment three times independently with HT29 cells and replaced the original data with a representative one showing 100% cell viability at the lowest concentration of each IT. Nonetheless, the new data did not alter the IC50, therapeutic index, and therapeutic window of ER-PE24 ITs in vitro, as shown in Table 2 of the original manuscript. As shown below, the new data was reflected in Figure 3A in the revised manuscript.

Comment#2-5) Fig. 4: Due to the higher affinity of ER/0.2-PE24 lower IT concentrations should have been used. 0.1 mg/kg seems to be effective AND safe. Since the affinity is 20 times higher compared to ER/21-PE24 mouse experiments should have been performed with 10 to 20 times lower concentrations. The concentration range used is not optimal to draw the conclusion that the therapeutic window of the first IT is smaller compared to the second. The experiment only shows higher toxicity of the one with higher affinity what should be expected.

Response) Thank you for your comment. According to Figure 3 and Table 2 of the original manuscript, affinity and potency are not entirely consistent in vitro since cell killing is affected both by affinity and receptor internalization efficiency. In in vivo experiments, Figures 4A and B also showed that ER/0.2-PE24 and ER/21-PE24 had similar antitumor activity at 0.1 mg/kg and 0.2 mg/kg, indicating that despite their 100-fold differences in EGFR affinity, there was no significant difference in their antitumor potency. In previous reports with ITs, ITs with a 10-fold increase in affinity did not show either corresponding improvement in internalization or a concomitant improvement in cytotoxic effects [Yu Cao et al (2011) Mol Cancer Ther, 11: 143-153; Yariv Mazor et al (2017), Scientific Reports, 7: 40098]. To identify the therapeutic window, we first determined the maximal tolerated doses (MTDs) of the two ITs at higher doses and then determined the antitumor activity at lower doses than the MTD. In Figure 4, ER/21-PE24 showed a 4-fold improvement in MTD compared to ER/0.2-PE24, indicating a 4-fold expansion of the therapeutic window.

Reviewer 3 Report

The current study designed and constructed several EGFR-targeted PE24 immunotoxins and analyzed their binding kinetics and affinities and their in-vitro and in-vivo potencies against EGFR-high and -low expression cell models. The research methodology and experimental designs are sound and data supports the logic and conclusion of this work. I offer several factors for further consideration by the authors in the hope of improving this manuscript and RIT design under investigation.

With an MW ~40 KDa, these molecules may have a very short half-life in circulation. How does the author approach the question of reducing the dosage to limit toxicity and maintain the plasma concentrations to fall within the identified therapeutical windows?

Do EGFR-overexpressing malignancies in humans exhibit consistent and comparable EGFR expression levels, as shown in A431? The 10-20-fold in the surface expression levels was not a huge difference to start with. If the EGFR expression levels differ among the patients, does that mean the binding moiety’s affinity also needs to be adjusted to ensure the therapeutic window?

As shown in this study, to increase the width of the therapeutic window and avoid the normal cells with constitutive EGFR expression, a lower affinity binding moiety and higher IC50 are preferred. On the other hand, in the experience of previous RITs development, higher dosages and more frequent dosages bring about the ADA issue against the PE domain, which will negate the RIT efficacy in most patients after 6-8 rounds of treatment. The two factors pull the design of the treatment regime in different directions, how does the author balance these factors?

Author Response

Please see the uploaded PDF file of the responses to your comments, which is prepared in color font for your convenience.

---------------------------------------

Responses to specific comments of reviewer #3

The current study designed and constructed several EGFR-targeted PE24 immunotoxins and analyzed their binding kinetics and affinities and their in-vitro and in-vivo potencies against EGFR-high and -low expression cell models. The research methodology and experimental designs are sound and data supports the logic and conclusion of this work. I offer several factors for further consideration by the authors in the hope of improving this manuscript and RIT design under investigation.

Comment#3-1) With an MW ~40 KDa, these molecules may have a very short half-life in circulation. How does the author approach the question of reducing the dosage to limit toxicity and maintain the plasma concentrations to fall within the identified therapeutical windows?

Response) Thank you for your comment. As you already mentioned, ITs based on antibody fragments (scFv, Fab) without Fc portion have molecular weight of ~50 – 72-kDa and a very short serum half-life (~ 10 – 30 min in mice) [Wei et al (2018) Proc Natl Acad Sci U S A 115: E3501-3508]. ER-PE24 with ~35 kDa molecular weight will also show a very short half-life. If injections are given more frequently at higher doses to overcome the short serum half-life, severe systemic toxicity will result. Accordingly, in our in vivo studies in the mouse model, we first measured the maximum tolerated dose (MTD) of the two ER-PE24 ITs in every other day i.v. injection, followed by determining the antitumor activity at doses lower than the MTD. As shown in Figure 4, the lower doses of ER/21-PE24 at 0.4, 0.2, and 0.1 mpk substantially inhibited the growth of tumors, showing ~79%, ~69%, and ~57% tumor growth inhibition at the end of treatment without noticeable adverse effects, respectively, compared with the PBS-treated controls. A similar approach can be used in clinical trials to find optimal regimens (dosing amount and schedule).

Comment#3-2) Do EGFR-overexpressing malignancies in humans exhibit consistent and comparable EGFR expression levels, as shown in A431? The 10-20-fold in the surface expression levels was not a huge difference to start with. If the EGFR expression levels differ among the patients, does that mean the binding moiety’s affinity also needs to be adjusted to ensure the therapeutic window?

Response) Thank you for pointing it out. A431 was used as a representative EGFR-overexpressing cell line [Merlino et al (1984) Science 224, 417-419]. As you mentioned, the expression levels of EGFR vary a lot among solid tumors [Kries (2019) Sci Rep 9:13564; Douillard et al. (2014) J Thorac Oncol 9:717-724], indicating the need to adjust the EGFR affinity of EGFR-targeted ITs based on EGFR expression levels on targeted tumors to maximize the therapeutic window while minimizing the on-target/off-tumor adverse effects.

In response to your comment, we edited the following sentence in [Discussion] (p. 12) of the revised manuscript as follows:

“It is known that many solid tumors overexpress EGFR on the cell surface, but there is some heterogeneity between and within solid tumors in terms of the levels of EGFR expression [33]. Our results highlight the importance of adjusting the EGFR affinity of EGFR-targeted ITs based on EGFR expression levels on targeted tumors to maximize the therapeutic window while minimizing the on-target/off-tumor adverse effects.”

Comment#3-3) As shown in this study, to increase the width of the therapeutic window and avoid the normal cells with constitutive EGFR expression, a lower affinity binding moiety and higher IC50 are preferred. On the other hand, in the experience of previous RITs development, higher dosages and more frequent dosages bring about the ADA issue against the PE domain, which will negate the RIT efficacy in most patients after 6-8 rounds of treatment. The two factors pull the design of the treatment regime in different directions, how does the author balance these factors?

Response) We agree with your comment. Due to the small size (~35 kDa), ER-PE24 IT has a short serum half-life mainly due to rapid clearance by the kidneys [Lei et al (2020) Proc Natl Acad Sci U S A 117:6086-6091]. However, more frequent dosages at high doses to overcome the short serum half-life will cause adverse effects, including ant-drug antibody (ADA) generation and severe renal toxicity. As we wrote in the original manuscript, we used a B-cell epitope removed 24-kDa PE fragment variant, LO10 PE24, to minimize the immunogenicity issue. We observed that at doses higher than MTD, both ER/0.2-PE24 and ER/21-PE24 caused tumor-bearing mice (TBM) death before significant body weight loss, indicating their acute renal toxicity [Lei et al (2020) Proc Natl Acad Sci U S A 117:6086-6091]. Accordingly, we speculate that extension of serum half-life of ER-PE24 ITs by conjugating polyethylene glycol [29] or reformatting into an Fc-containing IgG can improve therapeutic window [Zheng et al (2020) Mol Cancer Ther 19:812-821]. Even in the case of ITs with extended serum half-life, the tumor antigen targeting affinity should be adjusted depending on the expression levels on tumors and normal tissues to avoid on-target/off-tumor toxicity and to improve intratumoral penetration.

In response to your comment,

In response to your comment, we added the following sentences in [Introduction] (p. 2) of the revised manuscript as follows:

“As a model toxin, we used a de-immunized 24-kDa PE fragment variant, LO10 PE24 [13, 14], which contains the furin cleavage site in domain II (residues 274-284) and the domain III (395-613) of PE with a replacement of the C-terminal endoplasmic reticulum (ER) retention sequence 609REDLK613 with 609KDEL612.”

We also added a following paragraph in [Discussion] (p. 12) of the revised manuscript as follows:

“At doses higher than MTD, both ER/0.2-PE24 and ER/21-PE24 caused TBM death before significant body weight loss, indicating their acute renal toxicity [24]. ER-PE24 ITs as small as ~35 kDa appeared to undergo rapid renal filtration, resulting in fatal kidney damage, as observed with PE38-carrying SS1P IT [24]. To minimize acute renal toxicity by reducing renal filtration, ER-PE24 IT needs to be further engineered to increase serum half-life, for example by conjugation of polyethylene glycol [30] or reformatting into an Fc-containing IgG. Even in the case of ITs with extended serum half-lives, the tumor antigen targeting affinity should be tailored depending on differences in the expression levels between tumors and normal tissues to avoid on-target/off-tumor toxicity and to enhance intratumoral penetration [31].”

Round 2

Reviewer 2 Report

All queries have been adressed.